# Risk and protective factors for self-harm and suicide in children and adolescents: a systematic review and meta-analysis protocol

Dan Farbstein [1,2] Steve Lukito [3] Isabel Yorke,[3] Emma Wilson [4,5] Holly Crudgington [4,5] Omar El-Aalem,[6] Charlotte Cliffe [7] Nicol Bergou,[8] Lynn Itani,[9] Andy Owusu,[6] Rosemary Sedgwick [6] Nidhita Singh,[6] Anna Tarasenko,[10] Gavin Tucker,[6] Emma Woodhouse,[11,12] Mimi Suzuki,[13] Anna Louise Myerscough [13] Natalia Lopez Chemas,[13] Nadia Abdel-Halim,[13] Cinzia Del Giovane,[14,15] Sophie Epstein [16] Dennis Ougrin [3,17]

DF and SL contributed equally. SE and DO contributed equally.

SE and DO are joint senior authors.

For numbered affiliations see end of article.

**Correspondence to**
Dr Steve Lukito;
steve.lukito@kcl.ac.uk

## ABSTRACT

**Introduction** Self-harm and suicide are major public health concerns among children and adolescents. Many risk and protective factors for suicide and self-harm have been identified and reported in the literature. However, the capacity of these identified risk and protective factors to guide assessment and management is limited due to their great number. This protocol describes an ongoing systematic review and meta-analysis which aims to examine longitudinal studies of risk factors for self-harm and suicide in children and adolescents, to provide a comparison of the strengths of association of the various risk factors for self-harm and suicide and to shed light on those that require further investigation.

**Methods and analysis** We perform a systematic search of the literature using the databases EMBASE, PsycINFO, Medline, CINAHL and HMIC from inception up to 28 October 2020, and the search will be updated before the systematic review publication. Additionally, we will contact experts in the field, including principal investigators whose peer-reviewed publications are included in our systematic review as well as investigators from our extensive research network, and we will search the reference lists of relevant reviews to retrieve any articles that were not identified in our search. We will extract relevant data and present a narrative synthesis and combine the results in meta-analyses where there are sufficient data. We will assess the risk of bias for each study using the Newcastle–Ottawa Scale and present a summary of the quantity and the quality of the evidence for each risk or protective factor.

**Ethics and dissemination** Ethical approval will not be sought as this is a systematic review of the literature. Results will be published in mental health journals and presented at conferences focused on suicide prevention.

**PROSPERO registration number** CRD42021228212.

## STRENGTHS AND LIMITATIONS OF THIS STUDY

⇒ This systematic review will examine data about risk and protective factors for suicide and self-harm in children and adolescents only from longitudinal studies, thus avoiding issues of temporality that exist in cross-sectional studies.

⇒ We will perform a thorough and systematic search including several databases from health and mental health fields, searching reference lists of previous systematic reviews and contacting key authors in the field.

⇒ Using the broad definition of self-harm (as opposed to differentiating between suicide attempt and non-suicidal self-injury) may create challenges in synthesising the data as its definition in each study may be different, but on the other hand will prevent us from missing important evidence on the subject.

⇒ Examination of all identified risk and protective factors for self-harm and suicide in children and adolescents simultaneously in one systematic review will allow comparison of the importance of each factor.

⇒ This systematic review will only include studies published in the English language and therefore may overlook risk and protective factors that are unique to individual countries or parts of the world.

## INTRODUCTION

Suicide and self-harm, defined as any act of self-poisoning or self-injury, regardless of the motivation[1] among adolescents and young people, are of major public health concern. Acts of self-harm are known to be highly prevalent in adolescents. International community-based studies estimate that approximately 10%–30% of adolescents self-harm,[2 3] and 1%–10% report to have made a suicide attempt at least once in their life.[4–7] These numbers are even higher in specific at-risk populations, such as those with psychiatric disorders, most notably

depression and anxiety.[6] Most worryingly, self-harm and suicide attempts are some of the strongest predictive factors for completed, or death by, suicide,[8 9] which is now ranked as the second most common cause of death among those aged 10–24 years old, surpassed only by accidents.[10]

Due to their great importance, many efforts have been undertaken to prevent self-harm and suicide attempts among adolescents. For example, a large international study examining school-based intervention programmes for suicide prevention has shown that empowering youth and teaching coping skills result in decreased suicidal ideation and suicide attempts.[11] There have been recent advances in our understanding of precursors[12] and treatment[13] of self-harm in young people. However, the identification of adolescents at risk of self-harm and suicide attempt remains difficult. Various studies and meta-analyses have been undertaken in an attempt to shed light on specific risk factors for self-harm and suicide in this population. Pre-existing externalising (eg, conduct problems) and internalising symptoms (eg, depression or anxiety) of psychopathology have been demonstrated to be significant risk factors for suicidal behaviour.[14 15] Other risk factors include: sociodemographic factors, such as age,[16] gender, sexual orientation, transgender status and socioeconomic status[17 18]; stressful life events, most notably, childhood abuse[19 20]; familial factors, including familial psychopathology, family dysfunction, family conflict and perceived parental support[21–26]; school-related factors such as academic performance and school attendance[27 28]; biological factors such as genetic predisposition, changes in brain structure and function, and inflammatory status[29–31]; and social factors including peer relatedness, bullying perpetration and victimisation, and social media use.[16 26 32–36] Some studies have examined the relationship between self-harm or suicidal behaviour and specific emotions, such as hopefulness and self-esteem and emotional lability.[16 34 37 38]

While these studies are of great importance in elucidating the risk factors for self-harm and suicidal behaviour, their clinical utility is limited. Such a large number of risk factors is likely to be overwhelming for the clinician meeting an adolescent in the emergency department, and in the primary care setting, and thus not a helpful guide for the psychiatric examination. Additionally, many systematic reviews and meta-analyses take into account both cross-sectional studies and longitudinal studies.[18 27 36 39 40] Cross-sectional studies may be highly informative about the associations between various exposures and outcomes. However, risk or protective factors that are examined in cross-sectional studies cannot predict future events, due to the unclear temporal precedence between predictors and outcomes.

This protocol describes an ongoing systematic review and meta-analysis which will examine the evidence from longitudinal studies of all identified risk factors for self-harm and suicide in children and adolescents. This review will have two aims:

1. To provide summary strengths of association of the various risk factors for self-harm and suicide.
2. To identify risk factors for self-harm and suicide that are still debated in the literature, and thus to shed light on areas that are still in need of investigation.

## METHODS

Our systematic review protocol follows the Preferred Reporting Items for Systematic Review and Meta-Analysis Protocols (PRISMA-P) guidelines[41] (online supplemental file 1—PRISMA-P checklist). The reporting of the systematic review and meta-analysis resulting from this protocol will follow the PRISMA2020 guidelines,[42] and a completed PRISMA2020 checklist will be submitted with the publication. The protocol is registered on PROSPERO, the international register of systematic reviews (ID CRD42021228212). Any changes to the protocol will be recorded on PROSPERO.

### Eligibility criteria
#### Study population
Our target population is male or female children (ie, young persons aged up to 10 years) and adolescents (ie, young persons aged 11–17 years in early, mid and late adolescence). The population must be under the age of 18 years at the time of both exposure to the risk factor and outcome (self-harm or suicide). The decision to choose 18 years as the upper age limit was based on the fact that in most countries, this is the age at which people transition to adulthood in many aspects. First, in many countries, this is the age at which people complete school. Second, age 18 years is when people legally become adults. Third, age 18 years is the typical age at which the transition from child and adolescent to adult mental health services occurs. If a study includes both young persons under 18 years and young adults who are over 18 years, then the study will only be included if (1) the average age of participants is under 18 years or (2) more than 50% of the participants are under 18 years or (3) there is a subgroup analysis for participants in the study meeting the criterion (1) or (2).

#### Types of studies to be included
We will include only quantitative longitudinal studies, either prospective or retrospective. Thus, cross-sectional studies will be excluded. Retrospective studies that measure the existence and extent of risk factors using recall will be excluded, to avoid bias which tends to result in the exaggeration of the relationship between the risk factor and the outcome measured. As we consider many different risk/protective factors altogether, we do not set a threshold of eligibility for follow-up time in this study. Additionally, we will exclude intervention studies, qualitative studies, case reports or series, commentaries, editorials and conference abstracts. Only papers written in English language will be included.

### Exposure and outcome variables

We will include any study that examines the relationship between any risk or protective factor and self-harm or suicide. Studies included must examine self-harm or suicide as an outcome. There are many definitions and subdivisions for self-harm in the literature. As mentioned above, for our study, we decided to use the UK/European definition of self-harm, which is: any act of self-poisoning or self-injury carried out by a person, irrespective of their motivation.[43] This broad definition includes both (1) non-suicidal self-injury: self-inflicted destruction of body tissues without any suicidal intent,[44] and (2) suicide attempt: a non-fatal self-directed potentially injurious behaviour with any intent to die as a result of the behaviour.[45] Any type of ascertainment of the outcome variables is acceptable, including but not limited to self-report in questionnaires or interviews, collateral report, school or police records and medical files. As suicidal ideation is not our outcome of interest, studies that report on suicidal ideation as the only outcome will be excluded.

### Search methods

References will be extracted from the following databases according to the search strategy described below:
► Medline—Ovid platform
► PsycINFO—Ovid platform
► EMBASE—Ovid platform
► HMIC—Ovid platform
► CINAHL—EBSCO platform

Databases will be searched from inception, up to the present. The first wave of literature search was conducted to include studies published up to 28 October 2020, and the search will be updated before the systematic review publication.

For each of these databases, a search string was developed to include relevant keywords and synonyms (see 'search strategy' in online supplemental file 2). Examination of citation lists of previous relevant systematic reviews and meta-analyses will be conducted to retrieve any missed papers. Finally, experts in the field, which will include principal investigators whose peer-reviewed publications are included in our systematic review, as well as investigators from our extensive immediate research network,[35 46–48] will be contacted to identify any missed papers.

### Search strategy for electronic databases

Scoping searches were initially performed to refine the search strategy and to optimise the balance between sensitivity and specificity. Search strategies were subsequently developed to include keywords and specific thesaurus terms for each database to include the following concepts:
1. Self-harm and suicide.
2. Populations under 18 years.
3. Association terms (eg, relationship, risk factor, correlation, protective factors, etc).
4. Study designs.

To overcome linguistic variations, truncations and wildcards were employed when necessary. The list of keywords and thesaurus terms was based on previous systematic reviews about self-harm and suicide[27 35] and was discussed and finalised by a team of experts in the field of self-harm research in children and adolescents, tertiary-level and secondary-level service clinicians, and experts in the fields of systematic reviews and meta-analyses. The concept of 'study designs' was introduced in other systematic reviews in this field,[14] and thus was employed in our search as well. For this concept, we used the keywords and thesaurus terms recommended by the *British Medical Journal Best Practice* to identify cohort and case–control studies,[49] but expanded the search to ensure sensitivity.

### Reference selection and data extraction

Our research team includes 22 team members, comprising a mixture of researchers, clinicians and clinical academics, including several with expertise in self-harm and several with extensive experience of systematic reviews and meta-analyses including a senior statistician. Following removal of duplicate references, a two-staged screening of references is undertaken aided by a prespecified screening instruction document. The document, which details the inclusion and exclusion criteria, is devised as follows: after producing a first draft of the screening instructions, the entire team will screen 100 references with each member blinded to the decision of the other team members. The decisions will be discussed together in team meetings, and any ambiguity or lack of clarity in the screening instruction will be addressed and specific clarifications added to the instructions as necessary. This process will be repeated until a 90% concordance rate is reached among all team members.

Due to the anticipated large number of references and the constraint of time and resources to conduct this large-scale systematic review, once ≥90% concordance has been reached during the initial calibration process, remaining references will be divided equally between all team members. Following this process, 10% of all references will be screened by two independent reviewers and references will be equally divided into sets and be distributed among all team members. Every set of references will be screened by one screener from the team during the title/abstract screening stage, with a random sample of 10% of the references reviewed by a second screener, following similar approaches in previous large-scale systematic reviews.[50–52] Agreement rates for every pair of screeners will be recorded and reported as per cent agreement.[53] If the agreement rates between pairs of screeners were lower than 90% within the set of 10% randomly selected articles, all remaining articles in this set will be double-screened. Articles progressing to full-text screening will be retrieved and screened by a single screener, with a random sample of 10% reviewed by a second screener, again similar to previous approaches.[50–52] If we identify more than one study, which includes the same or overlapping samples, we will prioritise studies using the following

hierarchy: largest sample, longest follow-up, adjusted for highest number of confounders. All discarded items will be recorded, including a reason for exclusion at the full-text screening stage. These screening stages will be conducted using the online systematic review tool Rayyan (https://www.rayyan.ai/).

Data will be extracted using a unified data extraction form that will be drafted a priori and collaboratively with the team. This will be piloted with the team and then for pragmatic reasons (due to the anticipated number of included studies), data on study characteristics will be extracted by single reviewers, with a random subsample of 10% of papers will be audited by a second reviewer. All statistical data relating to study results will be extracted by two independent reviewers. The following data will be extracted: type of study, participant eligibility and recruitment method, participant description including age, gender and ethnicity, diagnosis if applicable, country, study dates, sample size, definitions and measures of outcomes, blinding, duration of follow-up, number and type of predictors, definition and method for measurement of candidate predictors (eg, categories may include demographics, psychopathology, family relationships, peer relationships, treatment history, etc), timing of predictor measurement, missing data and its handling, authors, year of publication, type of statistical analysis, summary statistics and summary findings.

With regard to the summary statistics, we will collect effect measures expressing the association between risk factors and outcomes, as reported in individual studies, which are likely to include ORs, HRs, risk ratios (RRs), mean difference (MD) and standardised MD with 95% CIs and significant p values, or SEs where appropriate, as well as which covariates have been adjusted for.

Any disagreement over reference eligibility, during the title/abstract and full-text screening stages, or data extraction will be discussed between the two screeners/reviewers. If a consensus cannot be reached, a third screener (DO) will adjudicate.

### Risk of bias assessment

The Newcastle–Ottawa Scale (NOS)[54] will be used to assess risk of bias in observational/non-randomised studies. The risk of bias assessment will be shared out among multiple members of the research team with pairs of reviewers working independently applying the NOS. Separate versions of NOS are available for assessing the risks of bias in case–control and cohort studies, based on (1) the selection of study participants, for example, their representativeness and ascertainment; (2) the comparability between participant groups, that is, cases/exposed cohort or controls/non-exposed cohort; and (3) the risk of bias associated with the exposures in cohort studies, that is, the ascertainment of exposures, comparability of exposure ascertainment in case and controls, and response rates; or outcomes in cohort studies, that is, the assessments of outcomes, follow-up periods and adequacy of follow-up of cohorts. Assessment of risk of bias will be

conducted separately for case–control and cohort studies using the appropriate NOS versions.

Certain items of the NOS require tailoring to the specific systematic review or meta-analysis being conducted. A version of the NOS adapted for relevance to this research question has been developed and agreed a priori by the research team. In particular, we determined the acceptable level of loss to follow-up and acceptable methods of ascertainment of the exposure and outcome variables. Missing data within individual studies (eg, due to attrition) will be reported and be taken into account as part of the quality assessment, and the level of bias it introduced into the findings will be considered for each study.

Finally, the discrepancies between two reviewers regarding a study's risk of bias ratings will be resolved through discussions, if necessary, in the presence of a third reviewer (DO) who will adjudicate if a consensus cannot be reached.

### Data presentation and statistical analysis

Where possible, a meta-analysis will be conducted to synthesise evidence. Alternatively, we will perform a narrative synthesis without meta-analysis using the quantitative data of the included studies.[55] Depending on the available data, synthesis of quantitative data may involve summarising estimates of effect (ie, strength of association in our context) in narratives or vote counting based on the direction of the effects.[56] We will calculate missing outcome data, if possible, from the available data within the article to enable consistent reporting or potentially to include them in the meta-analyses. If this is not possible, the findings will be reported within the narrative synthesis in qualitative manner. The findings of association between risk factors and self-harm outcomes will be grouped according to categories of outcomes (ie, self-harm or completed suicide). Then, for each outcome group, the association between the risk factors and the outcome will be grouped by similar risk factor concepts, in line with clinically meaningful categorisation from past literature, for instance, 'existing psychopathology', including internalising, externalising and general psychopathology; 'family relationships', including family dynamics, parenting styles and familial resources; 'demographic variables', including socioeconomic status, ethnicity, sexual orientation and gender identity; 'peer relationships', such as bullying and victimisation, ostracism and interactions via social media; and previous self-harm events. The relative strength of associations between all identified factors with the outcome will be discussed. Extracted data about participant sample characteristics, study methodologies and a summary of findings from each study will be presented in a table of characteristics of included studies.

Random-effect meta-analysis will be conducted, given the anticipated variation of sample characteristics and outcome measures across studies. There is likely to be variation in the statistical effect measures used across the literature; however, most studies are expected to compare

those with and without the exposure of interest when estimating their association with binary self-harm outcomes (ie, with or without self-harm), and would use ORs, HRs or RRs to index such association. ORs are expected to be the effect measure mostly used by any eligible case–control studies. Furthermore, the ORs can be easily interpreted clinically.[57 58] For these reasons, synthesis will be conducted using OR summary statistics to examine the pooled association between each predictor and self-harm. Where possible, other effect measures will be transformed to ORs for synthesis.

The meta-analyses will be conducted on findings from studies investigating similar exposure and outcome variables. If a study reports multiple effects of interest, then we will include all the available effects. If there is more than one reported effect in relation to the same exposure and outcome, the effect adjusted for the largest number of confounders will be included. We will illustrate the data using a forest plot, with 95% CI and p value reported. It is expected that there will be significant heterogeneity between studies. Thus, the $I^2$ statistics will be calculated to index statistical heterogeneity in the meta-analysis. Finally, publication bias will be assessed by funnel plots and Egger's test if there are at least 10 studies within the meta-analysis (Cochrane Handbook 13.3.5.4[59]).

For each exposure and outcome, an assessment of strength of evidence will be performed by applying the Grading of Recommendations Assessment, Development and Evaluation (GRADE[60]) approach adapted for non-randomised studies. We will use the GRADE tool to assess the quality of evidence, that is, the certainty of the estimate of association between individual risk factors and each self-harm outcome. The GRADE tool will only be applied following a meta-analysis, that is, when the pooled evidence of association between specific risk factors and a self-harm outcome could be obtained. We will use the 95% CI to assess precision for the GRADE tool. If the 95% CI does not cross the line of non-significance, the risk estimate will be considered to be sufficiently precise.

Subgroup analyses according to age (children, ie, young persons aged up to 10 years vs adolescents, that is, young persons aged 11–17 years), gender (female vs male) and country (high-income vs low/middle-income country) will be conducted if possible. The quantitative synthesis of findings will be based on meta-analyses of all samples collected in the study. However, to assess the influence of risk of bias of the individual studies in the quantitative synthesis, we also plan to conduct sensitivity meta-analyses including only studies deemed to have low risk of bias (see Hetrick et al[61] for a similar approach). Studies with low risk of bias are defined by a total NOS score ≥7 following past reports.[62 63]

## Patient and public involvement

Not applicable.

## DISCUSSION

This systematic review and meta-analysis will attempt to outline the risk and protective factors for self-harm and suicide among children and adolescents. A major strength of our work will be its focus on longitudinal studies. The exclusion of cross-sectional studies will highlight the risk factors and protective factors that predict self-harm and suicide, as opposed to those that co-occur with them.

Conducting this systematic review and meta-analysis presents various challenges. Although we have used broad search terms to increase the chance of identifying all relevant data, there remains a chance that data could be missed due to the wide range of different terminologies used in different settings. Additionally, it is possible that suicide or self-harm may not be reported in the title or abstract and is omitted from the keywords if it is not a major outcome in the study. To reduce the risk of missing significant data, in addition to database searches, we will examine the reference lists of previous systematic reviews and meta-analyses on the subject and consult experts in the field (ie, principal investigators of peer-reviewed papers included in this review as well as investigators from our research network) to identify additional papers. An additional challenge will be the synthesis of the data. Besides the variation in the level of bias in the studies that needs to be accounted for, methodology may differ significantly for the measurement of different risk and protective factors. For example, psychiatric symptoms may be measured using various methods: clinical evaluation, questionnaires and structured or semistructured interviews, each method with its unique features in terms of reliability and validity. These variations will need to be taken into account when synthesising data of different studies.

The primary goal of our work will be to describe the strength of association between various risk and protective factors and self-harm and suicide outcomes in children and adolescents. We hope this work will inform the clinician assessing a child or adolescent in the outpatient clinic or emergency department regarding important areas that need to be covered to optimally assess future self-harm and suicide risk. The need for future research in different areas, whether due to insufficient studies, poorly conducted studies, or the lack of use of similar terminology or methodology that would allow adequate data synthesis and conclusions, will also be highlighted.

## ETHICS AND DISSEMINATION

Ethical approval for this study is not needed, since it is based on the review of published literature only. The full protocol and the findings of this study will be published in peer-reviewed journals whose major audience are practitioners in child and adolescent mental health.

**Author affiliations**
[1]Child and Adolescent Psychiatry Unit, Psychiatric Division, Rambam Health Care Campus, Haifa, Israel

[2]Technion Israel Institute of Technology, The Ruth and Bruce Rappaport Faculty of Medicine, Haifa, Israel

[3]Department of Child & Adolescent Psychiatry, Institute of Psychiatry, Psychology and Neuroscience, King's College London, London, UK

[4]Economic and Social Research Council (ESRC) Centre for Society and Mental Health, King's College London, London, UK

[5]Department of Health Service and Population Research, Institute of Psychiatry, Psychology and Neuroscience, King's College London, London, UK

[6]South London and Maudsley NHS Foundation Trust, London, UK

[7]Department of Biostatistics & Health Informatics, Institute of Psychiatry, Psychology and Neuroscience, King's College London, London, UK

[8]Department of Psychosis Studies, Institute of Psychiatry, Psychology and Neuroscience, King's College London, London, UK

[9]Emirates Health Services, Maudsley Health, Al Amal Psychiatric Hospital, Dubai, UAE

[10]BrainPatch Ltd, London, UK

[11]Compass Psychological Services Ltd, London, UK

[12]Department of Forensic and Neurodevelopmental Sciences, Institute of Psychiatry, Psychology and Neuroscience, King's College London, London, UK

[13]Unit for Social and Community Psychiatry, Queen Mary University of London & East London NHS Foundation Trust, London, UK

[14]Department of Medical and Surgical Sciences for Children and Adults, University-Hospital of Modena and Reggio Emilia, Modena, Italy

[15]Institute of Primary Health Care (BIHAM), University of Bern, Bern, Switzerland

[16]Department of Psychological Medicine, Institute of Psychiatry, Psychology and Neuroscience, King's College London, London, UK

[17]Youth Resilience Research Unit, WHO Collaborating Centre for Mental Health Services Development, Queen Mary University of London, London, UK

**Contributors** DF and SL—drafting of manuscript, designing of study, search strategy, screening process and analysis. IY—designing of study, search strategy, screening process and analysis, critical revision and review of the protocol. EW, HC, OE-A, NB, CC, LI, AO, RS, NS, AT, GT and EW—designing of study, search strategy, screening process and analysis, revision and review of protocol. MS, ALM, NLC and NA-H—screening process and analysis, revision and review of protocol. CDG, SE and DO—designing of study, search strategy, screening process and analysis, critical revision and review of protocol.

**Funding** This work was supported by the joint UK Medical Research Foundation (MRF-058-0005-RG-OUGRI) and Medical Research Council (MRC) (MR/R004927/1) grant to DO, and the Daniel Turnberg Travel Fellowship (DTTFR12/1055) to DF. EW and HC were supported by the Economic and Social Research Council (ESRC) Centre for Society and Mental Health at King's College London (ES/S012567/1). SE was funded by an MRC Clinical Research Training Fellowship (MR/T001437/1) and previously received salary support from an MQ Data Science Award and from the Psychiatry Research Trust. This work was also supported by funding from the UK Department of Health via the National Institute for Health Research (NIHR) Biomedical Research Centre (BRC) for Mental Health at South London (grant number N/A) and the Maudsley National Health Service (NHS) Foundation Trust and the IoPPN, King's College London (grant number N/A).

**Disclaimer** The views expressed are those of the authors and not necessarily those of the NHS, the NIHR, MRC, MRF, ESRC or the Department of Health.

**Competing interests** None declared.

**Patient and public involvement** Patients and/or the public were not involved in the design, or conduct, or reporting, or dissemination plans of this research.

**Patient consent for publication** Not required.

**Provenance and peer review** Not commissioned; externally peer reviewed.

**ORCID iDs**
Dan Farbstein http://orcid.org/0000-0003-2585-9287
Steve Lukito http://orcid.org/0000-0002-0616-5055
Emma Wilson http://orcid.org/0000-0003-4413-8338
Holly Crudgington http://orcid.org/0000-0003-1048-4953
Charlotte Cliffe http://orcid.org/0000-0003-3906-0003
Rosemary Sedgwick http://orcid.org/0000-0002-6344-9353
Anna Louise Myerscough http://orcid.org/0000-0002-4758-669X
Sophie Epstein http://orcid.org/0000-0002-2118-908X
Dennis Ougrin http://orcid.org/0000-0003-1995-5408

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
