## [Reviewer comments · BMJ Open]

ARTICLE DETAILS

TITLE (PROVISIONAL)	Risk and Protective Factors for Self-Harm and Suicide in Children and Adolescents: A Systematic Review and Meta-Analysis Protocol
AUTHORS	Farbstein, Dan; Lukito, Steve; Yorke, Isabel; Wilson, Emma; Crudgington, Holly; El-Aalem, Omar; cliffe, charlotte; Bergou, Nicol; Itani, Lynn; Owusu, Andy; Sedgwick, Rosemary; Singh, Nidhita; Tarasenko, Anna; Tucker, Gavin; Woodhouse, Emma; Suzuki, Mimi; Myerscough, Anna; Lopez Chemas, Natalia; Abdel-Halim, Nadia; Del Giovane, Cinzia; Epstein, Sophie; Ougrin, Dennis

VERSION 1 – REVIEW

REVIEWER	Connor, Charlotte University of Warwick Warwick Medical School, Mental Health and Wellbeing
REVIEW RETURNED	30-Mar-2022

GENERAL COMMENTS	This is a useful and timely review and I will be interested to see your findings. However, there are a few amendments required before publication. Abstract: The authors do not elaborate on what they mean by 'experts in the field'. Are these academics or clinicians or others? How will these experts be contacted and how will they be chosen? Introduction: Please define what is meant by 'adolescence'. There are three stages of adolescences (early (11-14 years); mid (15-17 years) and late (18-21 years). The authors state that they will only be including studies involving young people under 18 years. Please be specific about this target group throughout the paper. Please also be specific about the gender of these young people as this is not stated. In terms of the studies, the authors state 'no lower limit date' for studies reviewed. Please specify what period this covers. Discussion: Only longitudinal studies are to be included in the review. Please define what is meant by this. Longitudinal studies can range from a few weeks to a few years and it is not clear whether the authors had specific range in mind.
--

REVIEWER	Tham, Su-Gwan University of Manchester, Institute of Brain, Behaviour and Mental Health
REVIEW RETURNED	22-May-2022

GENERAL COMMENTS

In this protocol, the authors propose to synthesise longitudinal studies to identify risk and protective factors and the magnitude of association with suicide and self-harm in children and young people.

The suggested search strategy is relatively comprehensive and includes controlled vocabularies and free text search terms, though I have made some suggestions to help streamline this. For the most part, it is broken down into concepts. An additional strength is that the authors propose to conduct assessment of risk of bias and the certainty of evidence.

My major queries relate to the proposed method and important information currently missing from the protocol which I detail first. The submitted reporting checklist has also not been completed as page numbers are not specified for each reporting item.

Suggested search strategy

- Line 2 – “self injurious behavior?r*” would suggest hyphenating.
- Lines 10 and 11 – autoaggress and automulti needs a hyphen between auto-[word] for accuracy as this will bring up auto-aggress and autoaggress.
- The wildcard symbol ? represents one character or none and is typically best used when there are spelling variants. It is unclear what the intention is when used at lines 16 and 17.
- There should be a subject heading for children and young people for Medline.
- Line 32 - ‘exp association/’ appears to be a generic subject heading. I would advise checking what this captures.
- Line 56 will capture both ‘follow-up’ and ‘follow up’ making lines 57 and 58 redundant.
- Line 70 and 71 - You have used this term as both a subject heading and free-text search term. I suggest using one or the other.
- Line 74 - According to your methodology, you do not intend to set any date limits.
- Line 88 - The syntax would be more comprehensive if lines 64-68 were incorporated with the first concept (i.e., lines 1-17).
- Line 89 - Again, add lines 70-71 to the second concept above to avoid this.
- It might be worth including the search terms ‘overdose’ and ‘student’ at the relevant parts of your search strategy.

Methods

- What reporting guidelines will you adhere to in your write-up?
- Will you include clinical or non-clinical population, or both?
- What will the authors do if a study includes both children and young people and adults? It isn't clear if such studies will be included or excluded.
- It should be noted in the eligibility criteria that studies published in languages other than English will be excluded.
- By ‘comments’ (line 51, page 17) – do you mean commentaries?
- Will you include or exclude papers where suicide ideation is examined?
- How will you initially identify experts in the field to identify missed papers? This should be stated.

	 • Line 25-27, page 18 – how many experts are in this team? • ‘Scoping searches were performed to refine the search strategy and optimize the balance between sensitivity and specificity’ (line 39-40, page 18) - this sentence should be near or at the top of this paragraph to better increase the readability and flow of the authors' writing. • Lines 50-51, page 18 – What level will the 100 references be screened at? De-duplication should be conducted prior to screening and this information is omitted. How and where will the authors deal with duplicates of search results? • I am unsure why the screening instructions need to be revised? The eligibility should be detailed and clear for anyone to be able to screen search results. Discussion is only generally required to resolve discrepancies in judgement, which is not mentioned. • What programme or software will be used to facilitate screening and distribution of references? • Given the size of the research team (17 members) and the review scope, I would expect a larger proportion of articles to be double screened. A strong rationale is required for a single team member to screen and extract data. • Line 9-10, page 19 - This doesn't follow the previous sentence. Why will agreement rates be reported when only a subsample of 10% of studies will be screened by a second independent reviewer? It is best to explain what will happen if there is disagreement among the 10% of studies that are double-screened. If indeed you are providing agreement rates, how will you calculate this? • How will discrepancies be resolved at title/abstract and at full-text stage? • Where will the data be extracted to? • Why will data be extracted by two independent reviewers if a potential meta-analysis will be conducted but not for the systematic review portion? Also, what data will be extracted for the potential meta-analysis? Specific detail is needed here. • Why are country and study dates part of participant characteristics? It seems better placed as part of study characteristics. As this review is examining self-harm and suicide, would psychiatric diagnoses be an important variable to extract? • Line 35, page 19 - examples of candidate predictors are required here. • How will the authors deal with potentially missing data from eligible studies? • How will you address publications that include the same sample? Risk of bias (quality) assessment • Risk of bias is not interchangeable with quality, especially if authors are using NOS and GRADE. Risk of bias is study limitations and quality of evidence is measuring the extent to which the estimated effect is accurate.
--	---

	 • Who will be carrying out risk of bias assessment? Is it the same reviewer who is screening and data extracting? • Why was the NOS tool chosen? Detail is needed re what the tool includes (e.g., examined domains) and the scoring system. • Lines 3-4, page 20 – potential adaptations of the NOS should ideally be stated already to avoid potential author biases and post-hoc decisions which may affect the rigour of the results. Data presentation and statistical analysis • Lines 13-14, page 20 – given that case studies and case series will be excluded, it would not be possible to do this? Please also consider use of the term 'participant' rather than 'subject.' • Line 17, page 20 – please define what this means for the reader. This type of information is not mentioned in your data extraction section nor mentioned as part of your criteria. • Line 33, page 20 - 'In each group' - does this mean single participant versus groups? This is not really feasible considering the study designs the authors are electing to include (e.g., longitudinal cohort studies) and exclude (e.g., case reports and series) • Line 25, page 20 - more detail is needed about the narrative synthesis. What type of narrative synthesis will be used and what are the specific steps that will be undertaken? Summarising based on risk factor will likely preclude the authors' ability to appropriately apply GRADE. GRADE appraises overall certainty of evidence per outcome measure. • There is no mention of odds ratios or any statistical data in your data extraction section. Also, why is odds ratio the chosen metric here? This implies the authors are comparing two groups based on the association between exposure and outcome. What groups are being compared here? Detail is required. • This section lacks sufficient detail. For example, how will effect sizes be pooled and transformed into a common metric? What happens if a study reports multiple effects of interest, how will the authors deal with this? Will they use all available effects or will they prioritise certain effects? It is not clear whether effects will be pooled based on risk/protective factor or outcome. How will authors deal with potential overlap of samples among studies? There is also no consideration of heterogeneity analyses nor how they intend to deal with potential identification of publication bias. Additionally, subgroup and sensitivity analyses have not been considered, which is a serious concern considering the large scope of this review. 'CI' should also be written out in full. Have the authors considered using the prediction interval too?
--	--

	 • How will you actually rank factors? • GRADE should only be conducted on an outcome basis only. The GRADE approach has not been adapted for non-randomised studies. It can be applied for both RCTs and observational studies - the latter is automatically degraded to 'moderate' certainty of evidence. • The strength of the evidence using the GRADE is not based on a group of studies, it is based on an outcome basis only. A total of 8 domains are used in GRADE, 5 of which are used to downgrade the certainty of evidence. Based on the detail provided in how you intend to deal with the synthesis, it does not suggest that using GRADE is a viable option. For example, how will the authors assess 'imprecision' as part of the GRADE process? Imprecision specifically examines the precision of the effect with its 95% CI for a particular outcome (e.g., attempted suicide). 'Inconsistency' cannot be measured either as there is insufficient detail in how the authors will be assessing heterogeneity (for both narrative and metaanalysis). It is not clear how the authors will examine such domains in GRADE and as it currently reads it will not be possible to apply the GRADE approach. In the absence of a meta-analysis, the narrative synthesis must incorporate processes such as transformation of effects into a common metric on an outcome basis for authors to judge imprecision. To assess inconsistency, study, participant and methodological characteristics all need to be considered for this domain to be judged appropriately. All these factors are critical as they influence the size and direction of the overall effect per outcome. • The proposal for why a score below 60% on the NOS will be included needs justification. It seems too arbitrary, particularly as there appears to be insufficient detail in delineating the scoring system of the NOS and the specific domains it assesses. Why are the authors considering filtering their results based on the NOS score but not GRADE? The authors will need to consider whether GRADE will be applied to all outcomes of interest or only those outcomes considered critical or important as per Cochrane standards. • More detail is required for potential subgroup analyses. How will authors manage with grouping e.g., age bands, gender and ethnicity? The authors need to consider if subgroup analyses will be conducted on an outcome basis or on a risk/protector factor basis. • Lines 20-22, page 21 - What synthesis will be employed? What are the steps of the synthesis? It is unclear if the authors are referring to the potential meta-analysis or narrative synthesis. Abstract
--	--

	 • Line 15, page 12 – ‘systematic review.’ Suggest using the broader term ‘evidence synthesis.’ As you propose to undertake both a systematic review and meta-analysis. • Line 25, page 12 - ‘we performed’ – in past tense. • Line 27, page 12 – should be ‘PsycINFO’ rather than ‘PsychINFO.’ • Line 27, page 12 – you note no lower date limit, but this contradicts the example search syntax (Line 74). Strengths and limitations • Lines 19-20, page 13 - citations tend to be part of the text, whilst reference refers to the full bibliographic information of citations. Reference lists is more accurate. • Line 34, page 13 - is it feasible or even possible to identify and include every risk and protective factor? Introduction • Line 19, page 14 - ‘suffering from mental disorders’ – please consider your phrasing of this. • Line 24, page 14 – consider the use of the term ‘completed suicide.’ For example, you could note ‘death by suicide’ instead. • Line 29, page 14 - specify further details of the type of accident if available e.g., road traffic accident. • Line 49, page 14 - ‘pre-existing externalizing and internalizing psychopathology’ – I’m not sure what this means. Perhaps consider rephrasing. • Lines 25-26, page 15 – this applies to other settings too such as primary care during a GP consultation. • Lines 29-31, page 15 – please cite examples of these. • Aim 1 refers to ‘predictive ability’. This is vague. Do you mean strength of the association? Predictive ability heavily implies a regression or correlational analysis will be carried out.
--	--

VERSION 1 – AUTHOR RESPONSE

Reviewer 1:

Comments to the Author:

This is a useful and timely review and I will be interested to see your findings. However, there are a few amendments required before publication.

Abstract: The authors do not elaborate on what they mean by ‘experts in the field’. Are these academics or clinicians or others? How will these experts be contacted and how will they be chosen?

These experts are both academics or clinicians who have published in the field of paediatric self-harm and suicide. We will contact principal investigators whose papers are included in our systematic review, as well as investigators from our extensive immediate research network. We have clarified this in the abstract (additional information is highlighted yellow): “...we will contact experts in the field, including principal investigators whose peer-reviewed publications are included in our systematic

review as well as investigators from our extensive research network”; and in the main text, section Search Methods last paragraph: “Finally, experts in the field, which will include principal investigators whose peer-reviewed publications are included in our systematic review, as well as investigators from our extensive immediate research network [e.g., 35, 46, 47, 48], will be contacted to identify any missed papers.”; and the Discussion Section (second paragraph): “... consult experts in the field (i.e., principal investigators of peer-reviewed papers included in this review as well as investigators from our research network)”

Introduction: Please define what is meant by 'adolescence'. There are three stages of adolescences (early (11-14 years); mid (15-17 years) and late (18-21 years)). The authors state that they will only be including studies involving young people under 18 years. Please be specific about this target group throughout the paper. Please also be specific about the gender of these young people as this is not stated.

Thank you. This lack of specification with respect to participant characteristics reflects the inclusivity of our literature search. We acknowledge the need for explicit reporting and have now added the required information in the section Eligibility Criteria, subsection Study Population: “Our target population is male or female children (i.e., young persons aged up to 10 years) and adolescents (i.e., young persons aged 11-17 years in early, mid and late adolescence).”

In terms of the studies, the authors state 'no lower limit date' for studies reviewed. Please specify what period this covers.

We have now clarified this in the main text section Search Methods: “databases will be searched from inception, up to the present. The first wave of literature search was conducted to include studies published up to 28 October 2020, and the search will be updated before the systematic review publication.” As we also have explained our initial systematic search of the literature has been conducted up to 28 October 2020. This search will be updated to include more recently published literature on the topic. A final search date will be stated in our publication.

Discussion: Only longitudinal studies are to be included in the review. Please define what is meant by this. Longitudinal studies can range from a few weeks to a few years, and it is not clear whether the authors had specific range in mind.

Our approach here has also been inclusive since we are considering many different risk/protective factors altogether and thus we have not specified a follow-up time. This is now clarified on section Eligibility Criteria, subsection 2 Types of Studies to be Included: “As we consider many different risk/protective factors altogether, we do not set a threshold of eligibility for follow-up time in this study.”

Reviewer 2:

In this protocol, the authors propose to synthesise longitudinal studies to identify risk and protective factors and the magnitude of association with suicide and self-harm in children and young people. The suggested search strategy is relatively comprehensive and includes controlled vocabularies and free text search terms, though I have made some suggestions to help streamline this. For the most part, it is broken down into concepts. An additional strength is that the authors propose to conduct assessment of risk of bias and the certainty of evidence.

My major queries relate to the proposed method and important information currently missing from the protocol which I detail first. The submitted reporting checklist has also not been completed as page numbers are not specified for each reporting item.

We have now attached a completed reporting checklist. We apologise for the oversight of submitting a blank checklist with our original manuscript.

Suggested search strategy

We thank the reviewer for these detailed suggestions. As the initial literature search and screening processes are now ongoing (Introduction, paragraph 4): “This protocol describes an ongoing systematic review and meta-analysis...”), we are not able to introduce search changes at the present wave of literature search (which include literature published until 28th October 2020). However, we can incorporate these suggestions, as necessary, in our next wave of literature search and will identify the remaining papers missed from the previous wave of search.

- Line 2 – “self injurious behavior?” would suggest hyphenating.

Thank you. Hyphenation is treated similarly as a blank space in Ovid®, the search platform we are using to conduct our literature search. Both terms (i.e., “self-injurious” and “self injurious”) will return identical search results.

- Lines 10 and 11 – autoaggress and automulti needs a hyphen between auto-[word] for accuracy as this will bring up auto-aggress and autoaggress.

We have re-run the search including and excluding the terms “auto-aggress*” and “auto-mutilate*” on 08.08.2022. From this exercise we retrieved 7 articles which were not retrieved using the original search strategy. Note that these articles may still be excluded at the screening stage. Our broad search terms appeared to have been sufficiently inclusive and have resulted in very few studies missed in the first place. We can identify papers specifically associated with the search terms “auto-aggress*” and “auto-mutilate*”, if necessary, and add them during the second wave of literature screening.

- The wildcard symbol ? represents one character or none and is typically best used when there are spelling variants. It is unclear what the intention is when used at lines 16 and 17.

Thank you. The reviewer was right to point out that the search terms “kill ???self” and “kill ???selves” do not function as we envisioned in the list of search term we have provided. We have since re-run the search adding and removing the term “kill himself”, “kill herself” and “kill themselves” on 08.08.2022 to assess the changes that these fully expanded terms would have added to the overall search. This exercise retrieved 2 articles only on top of what has been retrieved using the original search terms, and these articles may still be excluded at the screening stage. We will likely remove this term from the updated search terms.

- There should be a subject heading for children and young people for Medline.

Yes indeed. We have included those in lines 70 and 71.

- Line 32 - ‘exp association/’ appears to be a generic subject heading. I would advise checking what this captures.

Thank you. This line is indeed not necessary as it captures articles within the cognitive domain of learning. This term is a remnant from the early planning stage of the literature search that has been included mistakenly. We have re-run the search including and removing the term “exp association/”,

which retrieve the same number of articles. We thus do not envisage any false hits resulting with the inclusion of this line. However, we will remove this search when updating the literature search.

- Line 56 will capture both 'follow-up' and 'follow up' making lines 57 and 58 redundant.

The line 56 "follow-up" indeed will make line 57 "follow up" redundant. Thank you for noticing this typo. The line 58 "follow* up" is intended to capture phrases "followed up", "following up" in study abstracts associated with longitudinal study design. We intend to keep these terms as they are.

- Line 70 and 71 - You have used this term as both a subject heading and free-text search term. I suggest using one or the other.

Thank you. We believe that the MeSH terms for children and adolescents can enhance the search sensitivity and would keep them as they are.

- Line 74 - According to your methodology, you do not intend to set any date limits.

The limit that the reviewer refers to here pertains the term "exp epidemiologic methods/", not the whole search strategy. This was intended to capture primarily cohort studies in Medline as specified by the BMJ Best Practice guidance (<https://bestpractice.bmj.com/info/toolkit/learn-ebm/study-design-search-filters/> - the reference is included in the original manuscript).

- Line 88 - The syntax would be more comprehensive if lines 64-68 were incorporated with the first concept (i.e., lines 1-17).
- Line 89 - Again, add lines 70-71 to the second concept above to avoid this.

Yes. We have indeed separated the search conducted using the MeSH terms and using the key terms that we have come up ourselves. However, each set of MeSH terms and non-MeSH key terms associated with the same concept were combined using the appropriate Boolean operator OR (see lines 88-90), whereas combination across concepts such as those related to self-harm or suicide, children/adolescent are done with the Boolean operator AND (line 91). The grouping of MeSH and non-MeSH key terms by concepts as the Reviewer 2 suggested would not result in increased or reduced comprehensiveness of the retrieval of articles.

- It might be worth including the search terms 'overdose' and 'student' at the relevant parts of your search strategy.

Our scoping exercise suggests that the term "student" in combination with the term self-harm/suicide key terms identified predominantly studies about older student groups, such as medical or nursing students and undergraduate/graduate students. For this reason, we do not include the term in our final selection of key search. From the scoping exercise, we also found that the search term "overdose" identified overwhelmingly studies about accidental overdose or studies that do not differentiate accidental overdose from suicidal acts. For this reason, we have used instead the term self-poisoning, expressed on line 14 self-poison* on the search list.

Methods

- What reporting guidelines will you adhere to in your write-up?

We have now included this detail on the Methods section, paragraph 1: "The reporting of the systematic review and meta-analysis resulting from this protocol will follow the PRISMA2020 guidelines, and a completed PRISMA2020 checklist will be submitted with the publication." We have

also added page numbers to the reporting checklist for protocol of a systematic review and meta-analysis re-submitted with this review response.

- Will you include clinical or non-clinical population, or both?

We will include adolescents from both clinical and non-clinical populations. Clinical conditions are not an exclusion for our review.

- What will the authors do if a study includes both children and young people and adults? It isn't clear if such studies will be included or excluded.

Thank you for highlighting this lack of detail, we have now included this in the main text (section Eligibility Criteria, subsection Study Population): "If a study includes both young persons under 18 and young adults who are over 18, then the study will only be included if (1) the average age of participants is under 18 or (2) more than 50% of the participants are under 18 or (3) there is a subgroup analysis for participants in the study meeting the criteria (1) or (2)."

- It should be noted in the eligibility criteria that studies published in languages other than English will be excluded.

We have now stated this in the main text (section Eligibility Criteria, subsection Type of Studies to be Included): "Only papers written in English language will be included".

- By 'comments' (line 51, page 17) – do you mean commentaries?

Thank you for highlighting this typo, which has now been corrected.

- Will you include or exclude papers where suicide ideation is examined?

Papers which include suicidal ideation as the only outcome will be excluded. This has been clarified now (section Eligibility Criteria, subsection Type of Studies to be Included): "As suicidal ideation is not our primary outcome of interest, studies that report on suicidal ideation as the only outcome will be excluded."

- How will you initially identify experts in the field to identify missed papers? This should be stated.

These experts are both academics or clinicians who have published in the field of paediatric self-harm and suicide. We have clarified this in the abstract: "we will contact experts in the field, including principal investigators whose peer-reviewed publications are included in our systematic review as well as investigators from our extensive research network"; and in the main text (section Search Methods, last paragraph): "Finally, experts in the field, which will include principal investigators whose peer-reviewed publications are included in our systematic review, as well as investigators from our extensive immediate research network [e.g., 35, 46, 47, 48], will be contacted to identify any missed papers."; and the Discussion Section (second paragraph): "... consult experts in the field (i.e., principal investigators of peer-reviewed papers included in this review as well as investigators from our research network)"

- Line 25-27, page 18 – how many experts are in this team?

There are 17 team members at the start of the project, and our team has grown since to 22 members, who are a mixture of researchers, including 9 experienced systematic reviewers and 7 clinicians in mental health, several who have academic or clinical expertise in self-harm, and a senior statistician.

We have added further details of our team memberships (section Reference Selection and Data Extraction, paragraph 1): “Our research team consists of 22 members, comprising a mixture of researchers, clinicians and clinical academics, including several with expertise in self-harm and several with extensive experience of systematic reviews and meta-analyses including a senior statistician.”

- ‘Scoping searches were performed to refine the search strategy and optimize the balance between sensitivity and specificity’ (line 39-40, page 18) - this sentence should be near or at the top of this paragraph to better increase the readability and flow of the authors' writing.

Thank you. This has now been edited as suggested to improve the readability.

- Lines 50-51, page 18 – What level will the 100 references be screened at? De-duplication should be conducted prior to screening and this information is omitted. How and where will the authors deal with duplicates of search results?

We have now clarified these (section Reference Selection and Data Extraction, paragraph 1): “Following removal of duplicate references, a two-staged screening of references is undertaken aided by a pre-specified screening instruction document”.

- I am unsure why the screening instructions need to be revised? The eligibility should be detailed and clear for anyone to be able to screen search results. Discussion is only generally required to resolve discrepancies in judgement, which is not mentioned.

We have now clarified how the screening instructions for the study are refined (rather than revised) on the section Reference Selection and Data Extraction, paragraph 1: “The decisions will be discussed together in team meetings, and any ambiguity or lack of clarity in the screening instruction will be addressed and specific clarifications added to the instructions as necessary.”

- What programme or software will be used to facilitate screening and distribution of references?

We have clarified this on the section Reference Selection and Data Extraction, paragraph 2: “These screening stages will be conducted using the online systematic review tool Rayyan (<https://www.rayyan.ai/>)”

- Given the size of the research team (17 members) and the review scope, I would expect a larger proportion of articles to be double screened. A strong rationale is required for a single team member to screen and extract data.

Thank you. The approach that we have taken here are similar with those taken in previous large-scale systematic reviews (e.g., Baxter et al., 2018; Leaviss et al., 2020; Troy et al., 2022), chosen due to the constraint of time and resources for conducting this large-scale project. As explained above, the large number of our core team are experienced systematic reviewers, some involve in the teaching of the methodology, who are able to guide less experienced members of the team on the systematic review processes. We conduct an extensive calibration exercise to ensure that screening is done reliably with a piloted and refined screening instruction document. Finally, we have consulted our decision with a senior editor of the top journal in our field (JCPP) who advised that 10% double screening would be appropriate here in this context.

We provide some clarifications in the main text with respect to our screening (Reference Selection and Data Extraction, paragraph 2): “Due to the anticipated large number of references and the constraint of time and resources to conduct this large-scale systematic review, once ≥ 90%

concordance has been reached during the initial calibration process, remaining references will be divided equally between all team members. Following this process, 10% of all references will be screened by two independent reviewers and references will be equally divided into sets and be distributed among all team members. Every set of references will be screened by one screener from the team during the title/abstract screening stage, with a random sample of 10% of the references reviewed by a second screener, following similar approaches in previous large-scale systematic reviews [e.g., 50, 51, 52]. Agreement rates for every pair of screeners will be recorded and reported as percent agreement [53]. If the agreement rates between pairs of screeners were lower than 90% within the set of 10% randomly selected articles, all remaining articles in this set will be double-screened. Articles progressing to full-text screening will be retrieved and screened by a single screener, with a random sample of 10% reviewed by a second screener, again similar to previous approaches [50-52].”

We have further clarified about the data extraction (section Reference Selection and Data Extraction, paragraph 3): “Data will be extracted using a data extraction form developed collaboratively and a priori. This will be piloted with the team and then for pragmatic reasons (due to the anticipated number of included studies) data on study characteristics will be extracted by one reviewer, with a random subsample of 10% of papers will be audited by a second reviewer. All statistical data relating to study results will be extracted by two independent reviewers.”

- Line 9-10, page 19 - This doesn't follow the previous sentence. Why will agreement rates be reported when only a subsample of 10% of studies will be screened by a second independent reviewer? It is best to explain what will happen if there is disagreement among the 10% of studies that are double-screened. If indeed you are providing agreement rates, how will you calculate this?

Thank you. We have clarified as follows (section Reference Selection and Data Extraction, paragraph 3): “Agreement rates for every pair of screeners will be recorded and reported as percent agreement [53]. If the agreement rates between pairs of screeners were lower than 90% within the set of 10% randomly selected articles, all remaining articles in this set will be double-screened.” We expected that substantially smaller proportion of paper will be included in the review than initially identified. Therefore, agreement of rating would be calculated as percentage agreement (see e.g., Feinstein and Chiccetti, 1990).

- How will discrepancies be resolved at title/abstract and at full-text stage?

We have now stated this more explicitly by adding the following (section Reference Selection and Data Extraction, paragraph 5): “Any disagreement over reference eligibility, during the title/abstract and full-text screening stages, or during data extraction stage will be discussed between the two screeners/reviewers. If consensus cannot be reached, a third researcher (DO) will adjudicate.”

- Where will the data be extracted to?

“Data will be recorded using a unified data extraction form that will be drafted a priori and collaboratively with the team”. We have also now added that “Data extracted from the included studies will be presented in a table outlining their characteristics and summary findings.”

- Why will data be extracted by two independent reviewers if a potential meta-analysis will be conducted but not for the systematic review portion? Also, what data will be extracted for the potential meta-analysis? Specific detail is needed here.

Thank you. Our apologies for not being clear. This should have stated that any statistical data (whether for systematic review or meta-analysis) will be extracted by two reviewers, whereas data on study characteristics (less prone to extraction error) will be extracted by single reviewers with a random subsample of 10% audited by a second reviewer. We have clarified this above and in the main text as follows (section Reference Selection and Data Extraction, paragraph 3): “Data will be extracted using a data extraction form developed collaboratively and a priori. This will be piloted with the team and then for pragmatic reasons (due to the anticipated number of included studies), data on study characteristics will be extracted by one reviewer, with a random subsample of 10% of papers will be audited by a second reviewer. All statistical data relating to study results will be extracted by two independent reviewers.”

We have also further clarified what data will be extracted for the meta-analysis as follows (section Reference Selection and Data Extraction, paragraph 4):

“With regard to the summary statistics, we will collect effect measures expressing the association between risk factors and outcomes, as reported in individual studies, which are likely to include odds ratios (*OR*), hazard ratios (*HR*), risk ratios (*RR*), mean difference (*MD*) and standardised mean difference (*SMD*) with 95% confidence intervals (*CI*) and significance (*p*) values, or standard errors where appropriate, as well as which covariates have been adjusted for.”

- Why are country and study dates part of participant characteristics? It seems better placed as part of study characteristics. As this review is examining self-harm and suicide, would psychiatric diagnoses be an important variable to extract?

We believe that country and study dates would potentially induce a variation in the cohort of participants rather than the study design, and therefore will keep our description of country and study dates as part of the participant characteristics. We will indeed extract psychiatric diagnosis of the participants as appropriate, now clarified (section Reference Selection and Data Extraction, paragraph 3): “The following data will be extracted: ... diagnosis if applicable”

- Line 35, page 19 - examples of candidate predictors are required here.

Thank you. We have now added examples of candidate predictors as suggested (section Reference Selection and Data Extraction, paragraph 3): “... candidate predictors (e.g., categories may include demographics, psychopathology, family relationships, peer relationships, treatment history etc.)

- How will the authors deal with potentially missing data from eligible studies?

There will inevitably be different levels of missing data and loss to follow-up in each of the eligible studies. This will be accounted for as part of the quality assessment as the Newcastle Ottawa Scale has items on representativeness of the sample, and loss to follow up. There may also be missing data at a higher level such as missing outcome data, summary data or there may be studies missing from the literature due to publication bias. Each of these types of missing data will need to be managed in different ways. We have addressed the issue of publication bias in response to a subsequent comment. We have added the following detail to clarify how will manage the other types of missing data (section Risk of Bias Assessment, paragraph 3): “Missing data within individual studies (e.g., due to attrition), will be reported and be taken into account as part of the quality assessment, and the level of bias it introduced into the findings will be considered for each study.”

Furthermore, we discussed our approach on missing outcome data for each paper in the context of the synthesis (section Data presentation and statistical analysis, paragraph 1): “We will calculate missing outcome data in papers, if possible, from the available data within the article to enable consistent reporting or potentially to include them in the meta-analyses. If this is not possible, the findings will be reported within the narrative synthesis in qualitative manner.”

The issue of publication bias has also been addressed elsewhere (section Data presentation and statistical analysis, paragraph 3): “Finally, publication bias will be assessed by funnel plots and Egger’s test if there are at least ten studies within the meta-analysis (Cochrane Handbook 13.3.5.4; [59]).”

- How will you address publications that include the same sample?

Thank you. We have now addressed this (section Reference Selection and Data Extraction, paragraph 2): “If we identify more than one study which includes the same or overlapping samples, we will prioritise studies using the following hierarchy: largest sample, longest follow-up, adjusted for highest number of confounders”

Risk of bias (quality) assessment

- Risk of bias is not interchangeable with quality, especially if authors are using NOS and GRADE. Risk of bias is study limitations and quality of evidence is measuring the extent to which the estimated effect is accurate.

Thank you. We have now removed references that have mixed the two concepts. This includes: (1) rephrasing a sentence in the abstract “We will assess the risk of bias for each study using the Newcastle-Ottawa Scale”; (2) changing the section heading “Risk of bias (quality) assessment” to “Risk of bias assessment”; and (3) replacing quality with level of biases in the discussion section “Besides the variation in the level of biases...”

- Who will be carrying out risk of bias assessment? Is it the same reviewer who is screening and data extracting?

Thank you. We have added a clarification (section Risk of bias assessment, paragraph 1): “The risk of bias assessment will be shared out amongst multiple members of the research team with pairs of reviewers working independently applying the NOS.”

- Why was the NOS tool chosen? Detail is needed re what the tool includes (e.g., examined domains) and the scoring system.

We selected the NOS as it is a suitable tool for assessing risk of bias in observational studies. This is now more explicitly stated in the section Risk of Bias Assessment, paragraph 1: “The Newcastle-Ottawa Scale (NOS) [54] will be used to assess risk of bias in observational/non-randomised studies.” We have also provided further description about the NOS in the same paragraph: “Separate versions of NOS are available for assessing the risks of bias in case-control and cohort studies, based on (1) the selection of study participants, e.g., their representativeness and ascertainment; (2) the comparability between participant groups, i.e., cases/exposed cohort or controls/non-exposed cohort; and (3) the risk of bias associated with the exposures in cohort studies, i.e., the ascertainment of exposures, comparability of exposure ascertainment in case and controls, and response rates; or outcomes in cohort studies, i.e., the assessments of outcomes, follow-up periods, and adequacy of follow-up of cohorts. Assessment of risk of bias will be conducted separately for case-control and cohort studies using the appropriate NOS versions.”

- Lines 3-4, page 20 – potential adaptations of the NOS should ideally be stated already to avoid potential author biases and post-hoc decisions which may affect the rigour of the results.

We have now clarified this (section Risk of Bias Assessment, paragraph 2): “Certain items of the NOS require tailoring to the specific systematic review or meta-analysis being conducted. A version of the NOS scale adapted for relevance to this research question has been developed and agreed a priori by the research team. In particular, we determined the acceptable level of loss to follow-up and acceptable methods of ascertainment of the exposure and outcome variables”

Data presentation and statistical analysis

- Lines 13-14, page 20 – given that case studies and case series will be excluded, it would not be possible to do this? Please also consider use of the term ‘participant’ rather than ‘subject.’

Thank you. Apologies this was perhaps a confusing use of language. By ‘subject’ we meant ‘category’ of predictors, rather than subject to mean an individual participant. We hope that we can rectify this by the current revision (section Data presentation and statistical analysis, paragraph 1): “...The findings of association between risk factors and self-harm outcomes will be grouped according to categories of outcomes (i.e., self-harm or completed suicide). Then for each outcome group, the association between the risk factors and the outcome will be grouped by similar risk-factor concepts, in line with clinically meaningful categorisation from past literature, for instance ‘existing psychopathology’ including internalizing, externalizing, and general psychopathology; ‘family relationships’, including family dynamics, parenting styles and familial resources; by ‘demographic ...”

We have replaced all reference to “subjects” with “participants”

- Line 17, page 20 – please define what this means for the reader. This type of information is not mentioned in your data extraction section nor mentioned as part of your criteria.

We have now provided an explanation for the internalizing and externalizing symptoms of psychopathologies earlier on in (introduction, paragraph 2), where the terms appear for the first time: “Pre-existing externalizing (e.g., conduct problems) and internalizing symptoms (e.g., depression or anxiety) of psychopathology have been ...”. This has also addressed the reviewer’s comment below regarding line 49, page 14.

- Line 33, page 20 - ‘In each group’ - does this mean single participant versus groups? This is not really feasible considering the study designs the authors are electing to include (e.g., longitudinal cohort studies) and exclude (e.g., case reports and series)

Thank you. In this context, group means category of predictor and not participant group. We have removed this sentence for clarification.

- Line 25, page 20 - more detail is needed about the narrative synthesis. What type of narrative synthesis will be used and what are the specific steps that will be undertaken?

Thank you. This has now been further clarified (section Data presentation and statistical analysis, first paragraph): “Where possible a meta-analysis will be conducted to synthesise evidence. Alternatively, we will perform a narrative synthesis without meta-analysis using the quantitative data of the included studies [55]. Depending on the available data, synthesis of quantitative data may involve summarizing estimates of effect (i.e., strength of association in our context) using narratives, or vote counting based on the direction of the effects [56]. The findings of association between risk factors and self-harm outcomes will be grouped according to the categories of outcomes (i.e., self-harm or completed suicide). For each outcome group, the association between the risk factors and the outcome will be grouped by similar risk-factor concepts ...”

- Summarising based on risk factor will likely preclude the authors' ability to appropriately apply GRADE. GRADE appraises overall certainty of evidence per outcome measure.

We are conducting a systematic review and meta-analysis to synthesize findings regarding the association between risk factors and self-harm outcomes. For our study, we would use GRADE following a meta-analysis, to appraise the strength of evidence of the association between each risk factor and the outcome of self-harm.

- There is no mention of odds ratios or any statistical data in your data extraction section.

Thank you. We have now clarified this (section Reference Selection and Data Extraction, paragraph 3): “The following data will be extracted: ... type of statistical analysis, summary statistics and summary findings.”

In addition, in the next paragraph we described further the summary statistics that will be extracted: “With regard to the summary statistics, we will collect effect measures expressing the association between risk factors and outcomes, as reported in individual studies, which are likely to include odds ratios (*OR*), hazard ratios (*HR*), risk ratios (*RR*), mean difference (*MD*) and standardised mean difference (*SMD*) with 95% confidence intervals (*CI*) and significance (*p*) values, or standard errors where appropriate, as well as which covariates have been adjusted for.”

- Also, why is odds ratio the chosen metric here? This implies the authors are comparing two groups based on the association between exposure and outcome. What groups are being compared here? Detail is required.

We chose ORs as we are estimating the association between risk factors and a binary outcome (those with or without self-harm). The ORs is also an easily interpretable concept that is often used in clinical research thus would increase its accessibility for clinicians who are target audience of our work. We have now clarified this (section Data presentation and statistical analysis, paragraph 2): “Random-effect meta-analysis will be conducted, given the anticipated variation of sample characteristics and outcome measures across studies. There is likely to be variation in the statistical effect measures used across the literature, however, most studies are expected to compare those with and without the exposure of interest when estimating their association with binary self-harm outcomes (i.e., with or without self-harm), and would use *ORs*, *HRs* or *RRs* to index such association. *ORs* are expected to be the effect measure mostly used by any eligible case-control studies. Furthermore, the *ORs* can be easily interpreted clinically [57, 58]. For these reasons, synthesis will be conducted using *OR* summary statistics to examine the pooled association between each predictor and self-harm. Where possible, other effect measures will be transformed to *ORs* for synthesis.

- This section lacks sufficient detail. For example, how will effect sizes be pooled and transformed into a common metric?

This has been addressed now (in the section Data presentation and statistical analysis, paragraphs 2 & 3): “Random-effect meta-analysis will be conducted, given the anticipated variation of sample characteristics and outcome measures across studies.” Clarification is also added in the subsequent paragraph: “The meta-analyses will be conducted on findings from studies investigating similar exposure and outcome variables. ... We will illustrate the data using a forest plot, with 95% CI and p-value reported. It is expected that there will be significant heterogeneity between studies. Thus, the I^2 statistics will be calculated to index statistical heterogeneity in the meta-analysis. Finally, publication bias will be assessed by funnel plots and Egger’s test if there are at least ten studies within the meta-analysis (Cochrane Handbook 13.3.5.4; [59])”.

- What happens if a study reports multiple effects of interest, how will the authors deal with this? Will they use all available effects or will they prioritise certain effects?

Thank you, a clarification has now been added (in the section Data presentation and statistical analysis, paragraph 3): “If a study reports multiple effects of interest, then we will include all the available effects. If there is more than one reported effect in relation to the same exposure and outcome, the effect adjusted for the largest number of confounders will be included.”

- It is not clear whether effects will be pooled based on each category of risk/protective factor or outcome.

Thank you. This has now been addressed above (section Data presentation and statistical analysis, first paragraph): The findings of association between risk factors and self-harm outcomes would be grouped according to the categories of outcomes (i.e., self-harm or completed suicide). For each outcome group, the association between the risk factors and the outcome will be grouped by similar risk-factor concept ...

- How will authors deal with potential overlap of samples among studies?

Thank you. This has been addressed above (see section Reference Selection and Data Extraction, paragraph 2): “If we identify more than one study which includes the same or overlapping samples, we will prioritise studies using the following hierarchy: largest sample, longest follow-up, adjusted for highest number of confounders”

- There is also no consideration of heterogeneity analyses nor how they intend to deal with potential identification of publication bias.

Thank you, clarification is now added (Data presentation and statistical analysis, paragraph 4) : “It is expected that there will be significant heterogeneity between studies. The I^2 statistics will be calculated to index statistical heterogeneity in the meta-analysis”

It is also mentioned in the protocol that we will assess publication bias by using funnel plots and Egger’s test. We have also added a clarification “if there are at least ten studies within the meta-analysis (Cochrane Handbook 13.3.5.4) [59]”

- Additionally, subgroup and sensitivity analyses have not been considered, which is a serious concern considering the large scope of this review.

This was previously written in the main text, and it is now further clarified (section Reference Selection and Data Extraction, paragraph 5). We have decided to remove subgrouping analysis by ethnicity as we anticipate variation of ethnicity definition across countries. “Subgroup analyses according to age (children, i.e., young persons aged up to 10 years vs. adolescents, i.e., young persons aged 11-17 years), gender (female vs. male) and country (high- vs. low- and middle-income country) will be conducted if possible.”

- ‘CI’ should also be written out in full. Have the authors considered using the prediction interval too?

We have now written CI in full. “...95% confidence interval (CI) and significance (p) value.” No we don’t consider prediction interval, we believe these will only apply to a subset of studies.

- How will you actually rank factors?

Thank you. After discussion, we have decided that we will not rank factors given the difficulty in defining a cut-off point at which a factor would be considered clinically significant. We have removed text referring to the ranking of factors. The systematic review's aim would enable discussion of relative strengths of associations between predictors and outcomes. We have amended the protocol text accordingly.

- GRADE should only be conducted on an outcome basis only. The GRADE approach has not been adapted for non-randomised studies. It can be applied for both RCTs and observational studies - the latter is automatically degraded to 'moderate' certainty of evidence.

Thank you. We are aware of this, which makes it more difficult to use GRADE for observational studies. However, we are not aware of any other tool which assesses strength of evidence in reviews of observational studies. We have added some clarification (section Reference Selection and Data Extraction, paragraph 5): "We will use the GRADE tool to assess the quality of evidence, i.e., the certainty of the estimate of association between individual risk factors and each self-harm outcome. The GRADE tool will only be applied following a meta-analysis, i.e., when the pooled evidence of association between specific risk factors and a self-harm outcome could be obtained. We will use the 95% *CI* to assess precision for the GRADE tool. If the 95% *CI* does not cross the line of non-significance, the risk estimate will be considered to be sufficiently precise."

- The strength of the evidence using the GRADE is not based on a group of studies, it is based on an outcome basis only.

Thank you, this has been addressed as above. GRADE will be used to assess the strength of evidence pertaining the association between the risk factor and our self-harm outcome.

- A total of 8 domains are used in GRADE, 5 of which are used to downgrade the certainty of evidence. Based on the detail provided in how you intend to deal with the synthesis, it does not suggest that using GRADE is a viable option. For example, how will the authors assess 'imprecision' as part of the GRADE process? Imprecision specifically examines the precision of the effect with its 95% *CI* for a particular outcome (e.g., attempted suicide).

Thank you. As discussed above, we will use the GRADE tool following any meta-analyses and as described previously, the 95% confidence interval and *p* value will be included. Clarification have been written as above (Data presentation and statistical analysis, paragraph 4): "We will use the 95% *CI* to assess precision for the GRADE tool. If the 95% *CI* does not cross the line of non-significance, the risk estimate will be considered to be sufficiently precise."

- 'Inconsistency' cannot be measured either as there is insufficient detail in how the authors will be assessing heterogeneity (for both narrative and meta-analysis).?

Thank you. We have addressed this above. As we have also said, we would conduct GRADE assessment following a meta-analysis, if such synthesis was possible.

It is not clear how the authors will examine such domains in GRADE and as it currently reads it will not be possible to apply the GRADE approach.

Thank you. We have addressed this with the clarifications above.

- In the absence of a meta-analysis, the narrative synthesis must incorporate processes such as transformation of effects into a common metric on an outcome basis for authors to judge imprecision. To assess inconsistency, study, participant and methodological characteristics all need to be considered for this domain to be judged appropriately. All these factors are critical as they influence the size and direction of the overall effect per outcome.

Thank you, we believe that we have addressed this comment above.

- The proposal for why a score below 60% on the NOS will be included needs justification. It seems too arbitrary, particularly as there appears to be insufficient detail in delineating the scoring system of the NOS and the specific domains it assesses.

Thank you. We have now used a threshold of total score NOS of 7, that has been used in previous studies. We clarified this in the section Data Presentation and Statistical Analysis, last paragraph: “The quantitative synthesis of findings will be based on meta-analyses of all samples collected in the study. However, to assess the influence of risk of bias of the individual studies in the quantitative synthesis, we also plan to conduct sensitivity meta-analyses including only studies deemed to have low risk of bias (see e.g. [61] for a similar approach). Studies with low risk of bias are defined by a total NOS score ≥ 7 following past reports [e.g., 62, 63]”.

- Why are the authors considering filtering their results based on the NOS score but not GRADE? The authors will need to consider whether GRADE will be applied to all outcomes of interest or only those outcomes considered critical or important as per Cochrane standards.

As above, we have added a clarification to the use of the NOS score to assess the risk of bias of individual studies. We propose to use NOS scores to filter out studies associated with high risk of bias. We would like to see whether conducting meta-analysis using studies with lower risk of bias could result in a difference to the pooled outcome. We cannot use GRADE for this purpose because the tool assesses the strength of the evidence for the pooled findings, in this context the evidence of the association between risk factors and outcomes, and not the evidence derived from the individual studies.

- More detail is required for potential subgroup analyses. How will authors manage with grouping e.g., age bands, gender and ethnicity? The authors need to consider if subgroup analyses will be conducted on an outcome basis or on a risk/protector factor basis.

Thank you. This has been addressed above.

- Lines 20-22, page 21 - What synthesis will be employed? What are the steps of the synthesis? It is unclear if the authors are referring to the potential meta-analysis or narrative synthesis.

Thank you. We have now clarified this in the section Data Presentation and Statistical Analysis, last paragraph: “The quantitative synthesis of findings will be based on meta-analyses of all samples collected in the study. However, to assess the influence of risk of bias of the individual studies in the quantitative synthesis, we also plan to conduct sensitivity meta-analyses including individual studies deemed to have low risk of bias (see e.g. [61] for a similar approach). Studies with low risk of bias are defined by a total NOS score ≥ 7 following past reports [e.g., 62, 63]”.

Abstract

- Line 15, page 12 – ‘systematic review.’ Suggest using the broader term ‘evidence synthesis.’ As you propose to undertake both a systematic review and meta-analysis.

We have now added the word meta-analyses to the sentence.

- Line 25, page 12 - ‘we performed’ – in past tense.

We have rectified this and change to present tense. This is an ongoing systematic review, and we are currently conducting screening for the eligible studies.

- Line 27, page 12 – should be ‘PsycINFO’ rather than ‘PsychINFO.’

This typo has been corrected to PsycINFO.

- Line 27, page 12 – you note no lower date limit, but this contradicts the example search syntax (Line 74).

The limit that the reviewer refers to here pertains the term “exp epidemiologic methods/”, not the whole search strategy. This was intended to capture primarily cohort studies in Medline as specified by the BMJ Best Practice guidance, presumably because historically the more specific term to identify cohort studies did not exist within the year brackets.

Strengths and limitations

- Lines 19-20, page 13 - citations tend to be part of the text, whilst reference refers to the full bibliographic information of citations. Reference lists is more accurate.

We have now changed this. “We will perform a thorough and systematic search including several databases from health and mental health fields, searching for reference lists of previous systematic reviews and contacting key authors in the field.”

- Line 34, page 13 - is it feasible or even possible to identify and include every risk and protective factor?

Apologies, we did not mean to suggest that we could examine every possible risk/protective factors per se, rather we want to examine all risk/protective factors that we have identified through the systematic review. We have clarified as follows: “Examination of all identified risk and protective factors for self-harm and suicide in children and adolescents simultaneously in one systematic review will allow comparison of the importance of each factor.”

Introduction

- Line 19, page 14 - ‘suffering from mental disorders’ – please consider your phrasing of this.

We have now changed this phrase to: “These numbers are higher in specific at-risk populations, such as those with psychiatric disorders, most notably depression and anxiety”.

- Line 24, page 14 – consider the use of the term ‘completed suicide.’ For example, you could note ‘death by suicide’ instead.

We have added this suggested explanatory phrase in addition to the term “completed suicide” which is frequently used in the literature of this topic. “Most worryingly, self-harm ... strongest predictive factors for completed, or death by, suicide [8, 9]”

- Line 29, page 14 - specify further details of the type of accident if available e.g., road traffic accident.

We have now clarified this, using the same terminology included in the article cited. “Most worryingly, self-harm ..., which is now ranked as the second most common cause of death among 10-24 year olds, **surpassed only by traffic accidents**”

- Line 49, page 14 - ‘pre-existing externalizing and internalizing psychopathology’ – I’m not sure what this means. Perhaps consider rephrasing.

We have now included some clarifications as follows: “Pre-existing externalizing (e.g., conduct problems) and internalizing symptoms (e.g., depression or anxiety) of psychopathology have been ...”

- Lines 25-26, page 15 – this applies to other settings too such as primary care during a GP consultation.

We have now added this in the text, both in the introduction and discussion sections “Such a large number of risk factors is likely to be overwhelming for the clinician meeting an adolescent in the emergency department, or in the primary care setting, and thus not a helpful guide for the psychiatric examination”; and “We hope this work will inform the clinician assessing a child or adolescent in the outpatient clinic or the emergency department”

- Lines 29-31, page 15 – please cite examples of these.

We have now added cited examples.

- Aim 1 refers to ‘predictive ability’. This is vague. Do you mean strength of the association? Predictive ability heavily implies a regression or correlational analysis will be carried out.

We have changed the wording in this sentence to clarify our meaning “To provide a comparison of the strengths of association of the various risk factors or self-harm and suicide.”

Baxter, S., et al., The effects of integrated care: a systematic review of UK and international evidence. *BMC Health Serv Res*, 2018. 18(1): p. 350.

Leaviss, J., et al., Behavioural modification interventions for medically unexplained symptoms in primary care: systematic reviews and economic evaluation. *Health Technol Assess*, 2020. 24(46): p. 1-490.

Troy, D., et al., What is the impact of structural and cultural factors and interventions within educational settings on promoting positive mental health and preventing poor mental health: a systematic review. *BMC Public Health*, 2022. 22(1): p. 524.